# Clinical and Molecular Epidemiology of Hemorrhagic Fever with Renal Syndrome Caused by Orthohantaviruses in Xiangyun County, Dali Prefecture, Yunnan Province, China

**DOI:** 10.3390/vaccines11091477

**Published:** 2023-09-12

**Authors:** Hao Huang, Meng Fu, Peiyu Han, Hongmin Yin, Zi Yang, Yichen Kong, Bo Wang, Xinglou Yang, Tilian Ren, Yunzhi Zhang

**Affiliations:** 1Yunnan Key Laboratory of Screening and Research on Anti-Pathogenic Plant Resources from Western Yunnan, Key Laboratory for Cross-Border Control and Quarantine of Zoonoses in Universities of Yunnan Province, Institute of Preventive Medicine, School of Public Health, Dali University, Dali 671000, China; hh18760954826@outlook.com (H.H.); hanpeiyu1511@gmail.com (P.H.); 13099916216@163.com (H.Y.); ziy0617@163.com (Z.Y.); kycgreentime@outlook.com (Y.K.); 2School of Public Health, Kunming Medical University, Kunming 650000, China; lynx5923@outlook.com; 3Department of Biomedical Sciences and Pathobiology, Virginia-Maryland College of Veterinary Medicine, Virginia Polytechnic Institute and State University, Blacksburg, VA 24061, USA; bowang@vt.edu; 4Yunnan Key Laboratory of Biodiversity Information, Kunming Institute of Zoology, Chinese Academy of Sciences, Kunming 650023, China; yangxinglou@mail.kiz.ac.cn; 5Department of Infection, People’s Hospital of Xiangyun County, Dali 671000, China

**Keywords:** renal syndrome hemorrhagic fever, clinical features, orthohantavirus, genotyping, genetic diversity, zoonotic transmission

## Abstract

Hemorrhagic fever with renal syndrome (HFRS) is a zoonotic disease transmitted by several rodent species. We obtained clinical data of HFRS patients from the medical records of the People’s Hospital of Xiangyun County in Dali Prefecture from July 2019 to August 2021. We collected epidemiological data of HFRS patients through interviews and investigated host animals using the night clip or night cage method. We systematically performed epidemiological analyses of patients and host animals. The differences in the presence of rodent activity at home (χ^2^ = 8.75, *p* = 0.031 < 0.05), of rodent-proof equipment in the food (χ^2^ = 9.19, *p* = 0.025 < 0.05), and of rodents or rodent excrement in the workplace (χ^2^ = 10.35, *p* = 0.014 < 0.05) were statistically different in the four clinical types, including mild, medium, severe, and critical HFRS-associated diseases. Furthermore, we conducted molecular detection of orthohantavirus in host animals. The total orthohantavirus infection rate of rodents was 2.72% (9/331); the specific infection rate of specific animal species was 6.10% (5/82) for the *Apodemus chevrieri*, 100% (1/1) for the *Rattus nitidus*, 3.77% (2/53) for the *Rattus norvegicus*, and 12.50% (1/8) for the *Crocidura dracula*. In this study, a total of 21 strains of orthohantavirus were detected in patients and rodents. The 12 orthohantavirus strains from patients showed a closer relationship with Seoul orthohantavirus (SEOOV) L0199, DLR2, and GZRn60 strains; the six orthohantavirus strains from *Rattus norvegicus* and *Apodemus chevrieri* were closely related to SEOOV GZRn60 strain. One strain (XYRn163) from *Rattus norvegicus* and one strain (XYR.nitidus97) from *Rattus nitidus* were closely related to SEOOV DLR2 strain; the orthohantavirus strain from *Crocidura dracula* was closely related to the Luxi orthohantavirus (LUXV) LX309 strain. In conclusion, patients with HFRS in Xuangyun County of Dali Prefecture are predominantly affected by SEOOV, with multiple genotypes of orthohantavirus in host animals, and, most importantly, these orthohantavirus strains constantly demonstrated zoonotic risk in humans.

## 1. Introduction

The latest report of the International Committee on Taxonomy of Viruses (ICTV) elevates the original genus *Hantavirus* to the family *Hantaviridae* with four subfamilies, seven genera, and fifty-three species [1,2]. The hantavirus members in the subfamily *Mammantavirinae* are hosted by bats, moles, shrews, and rodents, and are divided into four genera: *Loanvirus*, *Mobatvirus*, *Thottimvirus*, and *Orthohantavirus*, among which only *Orthohantavirus* with rodents as hosts causes human diseases. Thus far, 10 of the total 38 species in the genus *Orthohantavirus* lead to human infections, causing two diseases: hemorrhagic fever with renal syndrome (HFRS), mainly distributed in Eurasia, and hantavirus pulmonary syndrome (HPS), mainly distributed in the American continent [3]. Hantaan orthohantavirus (HTNOV), Seoul orthohantavirus (SEOOV), Puumala orthohantavirus (PUUOV), and Dobrava–Belgrade orthohantavirus (DOBOV) cause HFRS [4]; in contrast, Sin Nombre orthohantavirus (SNOV), Andes orthohantavirus (ANDOV), Black Creek Canal orthohantavirus (BCCV), Bayou orthohantavirus (BAYOV), Choclo orthohantavirus (CHOOV), and Laguna Negra orthohantavirus (LANOV) cause HPS [5,6,7,8,9]. Orthohantavirus is a segmented, single-stranded, negative-stranded RNA virus with an envelope. The genome is divided into three segments: large segment (L), medium segment (M), and small segment (S). L fragment encodes RNA-dependent RNA polymerase (RdRp), which provides raw materials for viral genomic RNA replication and mRNA synthesis; the M fragment encodes the precursor proteins of viral envelope glycoproteins (Gn and Gc), and the S fragment encodes the nucleocapsid protein (NP) [10].

Genetically diversified orthohantavirus strains isolated from rodents, humans, and other hosts revealed the presence of at least 10 HTNOV subtypes and 6 SEOOV subtypes with significant geographic aggregation in different regions of China [11]. Previous studies have shown that each orthohantavirus genotype has a specific host for transmission and has co-evolved with the host [12]. For example, HTNOV and SEOOV are mainly carried by the Striped Field Mouse (*Apodemus agrarius*) and Norway rat (*Rattus norvegicus*), respectively [13]. Moreover, previous studies indicated that SEOOV was transmitted from *R. norvegicus* to *A. agrarius,* and HTNOV was also detected in *R. norvegicus* in the Shandong Province of China [14,15]. In addition, HTNOV was present in both species (*R. norvegicus* and *A. agrarius*) in Jiangsu Province, and SEOOV was also present in voles in Zhejiang Province, suggesting host spillover and host switching between different genotypes of orthohantavirus. Genetic variation in orthohantavirus is a result of host adaptation to environmental changes, and genetic diversity in orthohantavirus is the result of cross-species transmission, virus-host coevolution, and host migration, which may be an essential driver of orthohantavirus evolution and species formation [12,16]. In our previous work in 2011, we identified a novel orthohantavirus in the red-backed vole (*Eothenomys miletus*) in Luxi County of Yunnan Province (designated LUXV), which now has been assigned as the new orthohantavirus species *Luxi orthohantavirus* [1,17].

Orthohantaviruses can cause two types of human infections: hemorrhagic fever with renal syndrome (HFRS) and hantavirus pulmonary syndrome (HPS). In China, HFRS is mainly induced by HTNOV and SEOOV [18]. Among them, HTNOV causes severe HFRS, primarily hosted by *Apodemus agrarius*, which tends to live in agricultural areas, waterside meadows, forests, and logging [19,20]; SEOOV causes mild to moderate HFRS, and *Rattus norvegicus* is considered the primary host [21]. Europe and Asia are the central regions where HFRS is prevalent, with China accounting for approximately 90% of global cases. The incidence of HFRS has a certain seasonality, of which spring and winter are the highest, and the number of male cases is three times that of women, of which 87.3% are patients aged between 15 to 60, and the proportion of occupations as farmers is the largest, around 70% [22]. Notably, Dali Prefecture is the region with the highest incidence of HFRS, followed by Kunming City and Chuxiong Prefecture in Yunnan Province, southwest China [23]. In this study, we comprehensively investigated the clinical characteristics of HFRS patients and the molecular epidemiology of orthohantavirus in host animals from Xiangyun County, Dali Prefecture, Yunnan Province.

## 2. Materials and Methods

### 2.1. Statement of Medical Ethics

The relevant materials involving human and animal biomedical research were reviewed by the Medical Ethics Committee of Dali University, and it is considered that the project conforms to medical ethics (ethics certificate number: DLDXLL2020007), and it agreed to collect relevant samples and data.

### 2.2. Routine Blood Tests

2 mL anticoagulated venous blood was collected from patients. It is necessary to use sterile blood collection needles and correct disinfection during blood collection. Then the collected blood samples were automatically analyzed with a hematology analyzer (SYSMEX, XE-2100, Kobe, Japan), including red blood cell count, white blood cell count, platelet count, red blood cell specific volume, average red blood cell volume, average red blood cell hemoglobin content, and other indicators.

### 2.3. Urine Protein Test

In this study, qualitative urine protein testing was used, and the dry chemical method was used to detect the pH of urine in the initial test. Sulfosalicylic acid method was used to retest for positive results in the initial test. According to the results of the test strip and the different degrees of turbidity, sedimentation, and coagulation of the reaction liquid, the urine protein content is screened and roughly estimated.

### 2.4. Detection of Orthohantavirus IgG/IgM Antibodies

The *Orthohantavirus* IgM/IgG antibodies were detected by colloidal gold kit (BOSON BIOTECH, Xiamen, China). Laboratory operations were performed according to the instructions, which are as follows: leave the test card for 30 min, balance to room temperature, tear the aluminum foil bag, take out the test card, and lay it flat on the operating table. The control line C in the colloidal gold detection card is mouse anti-human *Orthohantavirus* polyclonal antibody, T1 is anti-human *Orthohantavirus* IgM antibody, and T2 is anti-human *Orthohantavirus* IgG antibody. Use a sampler to add 10 µL of the serum to be tested into the sampling hole, and after the specimen has penetrated the sample pad, add 2 drops (or 100 µL) of the specimen diluent. The test card only shows negative C control line; C control line and T1 IgM antibody line were IgM positive. C control line and T2 IgG antibody line were both IgG positive. If the C control line does not appear, this experiment is invalid. Test results within 15–20 min; more than 30 min is required to determine if the effect is invalid.

### 2.5. Description of Study Cases of HFRS Patients

HFRS is divided into clinically diagnosed cases (CDC) and laboratory-diagnosed cases (LDC). CDC is defined according to Chinese “WS 278-2008 Diagnostic Criteria for Epidemic Hemorrhagic Fever”. LDC is ascertained with the positive IgM/IgG against orthohantavirus by colloidal gold immunoassay of the patients with HFRS who were admitted to the Department of Infectious Diseases of the 9th Affiliated Hospital of Dali University (People’s Hospital of Xiangyun County of Dali Prefecture in Yunnan Province) from July 2019 to August 2021. The cases were classified into four clinical types: (1) Light: body temperature below 39 °C, mild symptoms of poisoning, no other bleeding phenomena except bleeding points, light kidney damage, no shock and oliguria; (2) Medium: body temperature 39–40 °C, severe symptoms of poisoning, obvious bulbar conjunctival edema, systolic blood pressure less than 90 mmHg or pulse pressure less than 30 mmHg during the course of the disease, with obvious bleeding and oliguria, urine protein (+++); (3) Severe: body temperature > 40 °C, severe poisoning symptoms and exudation signs, toxic psychiatric symptoms, and shock, skin ecchymosis and luminal bleeding, shock and kidney damage severe, oliguria, lasting less than 5 days or anuria within 2 days; (4) Critical disease: one of the conditions in which refractory shock, bleeding from vital organs, oliguria for more than 5 days or anuria for more than 2 days, heart failure, pulmonary edema, cerebral edema, cerebral hemorrhage, or cerebral herniation occurs on the basis of severe secondary infection.

### 2.6. HFRS Patient Specimens Collection

The whole blood and sera of patients diagnosed with HFRS from July 2019 to August 2021 were collected in the 9th Affiliated Hospital of Dali University. Whole blood samples were collected using vacuum blood collection tubes without anticoagulant substances. The blood samples were centrifuged, the upper serum was stored in cryopreservation tubes, and the liquid nitrogen preservation was returned to the laboratory and stored in a refrigerator at −80 °C for examination.

### 2.7. Medical Record Information Collection

Clinical data of the cases included general information about the patients, i.e., name, gender, age, ethnicity, occupation, and home address (current address); onset of illness, including date of onset, date of admission, admission diagnosis, date of discharge, discharge diagnosis, and clinical outcome; and clinical signs of the patients on admission and laboratory test results. According to the case investigation data in the “Surveillance Program of Hemorrhagic Fever with Renal Syndrome” of the Chinese Center for Diseases Control and Prevention, the epidemiological investigation was conducted on HFRS patients.

### 2.8. Host Animal Survey

From October 2020 to August 2021, the host animal surveys were conducted in nine residential areas and farming fields in Xiangyun County of Dali Prefecture, Yunnan Province. A host animal survey was performed by the night clip or night cage methods, using rat cage or rat clip as tools and homemade fried ham sausage as bait. Five rat cages were placed in each villager’s house, and a total of 100 rat cages were placed every day; rat traps were placed at five-meter intervals on the cultivated land. After the rodents captured at the scene were numbered, classified, and identified by morphology, two copies of liver and lung tissues were dissected, placed in cryopreservation tubes, stored with liquid nitrogen, transported in liquid nitrogen tanks, and returned to the laboratory for orthohantavirus detection.

### 2.9. Viral Nucleic Acid Extraction and Orthohantavirus Detection

Orthohantavirus nucleic acids were extracted from host animal liver and lung tissues as well as from patient blood samples using the QIAamp RNA kit (Qiagen, Hilden, Germany) according to the manufacturer’s instructions. The viral RNA is stored in a −80 °C freezer until use. RT-PCR amplification of the L-gene fragment of orthohantavirus was performed on the extracted RNA with the orthohantavirus universal primer: HAN-L-F1: 5′-ATGTAYGTBAGTGCWGATGC-3′, HAN-L-R1: 5′-AACCADTCWGTYCCRTCAT-C-3′; HAN-L-F2: 5′-TGCWGATGCHACIAARTGGTC-3′, and HAN-L-R2: 5′-GCRTCRTCWGARTGRTGDGCAA-3′ [24], using the FastKing One-Step RT-PCR Kit (TIANGEN BIOTECH, Beijing, China) and Premix Ex Taq polymerase (TaKaRa Bio Inc, Shiga, Japan) according to the manufacturer’s instructions. Nested RT-PCR was performed with a total reaction system of 25 μL. Reaction conditions: reverse transcription at 42 °C for 30 min, pre-denaturation at 95 °C for 3 min, followed by denaturation at 94 °C for 30 s, annealing at 50 °C for 30 s, extension at 72 °C for 1 min for 35 cycles, followed by supplemental extension at 72 °C for 30 s, and finally cooling at 12 °C for 1 min to be used. PCR products were extracted on a 1.2% agarose gel for electrophoretic analysis and observed in a UV gel imager for the presence or absence of target bands of specific molecular weight size (about 412 bp), and gel images were saved.

### 2.10. Species Identification of Host Animals

Nucleic acids extracted from the liver and lung tissues of host animals were taken. The cytochrome b (Cyt b) gene was amplified using universal primers MamCybF: 5′-ATGATATGAAAAACCATCGTTG-3′ and MamCybR: 5′-TTTCCNTTTCTGGTTTACAAGAC-3′ as described previously [25,26]. The amplicons were sent to the company (Shanghai Biotech, Shanghai, China) for sequencing and compared with references in the NCBI database to determine the species of the host animal.

### 2.11. Sequence and Phylogenetic Analysis of Orthohantavirus

The positive products from orthohantavirus-specific RT-PCR were purified by gel cutting using MinElute Gel Extraction Kit (Qiagen, Hilden, Germany) and sent to the company (Shanghai Biotech, Shanghai, China) for sequencing with both forward and reverse primers. The sequencing results were assembled and analyzed using the Seqman and EditSeq models in DNASTAR Lasergene Bioinformatics Software version 17, and then the orthohantavirus sequences were analyzed for identity comparison using the Megalign model in the software. Afterward, the newly detected orthohantavirus sequences in this study were compared with the existing orthohantavirus sequences in the GenBank database, and the sequence alignment was performed using Clustal X. Sequence similarity heatmaps are plotted using R and its package. On the official website of R (https://www.r-project.org/ (accessed on 8 March 2023)), select the corresponding operating system (Windows 10 to download the latest installation package version, and follow the installation wizard to complete the software installation. By website (https://www.rstudio.com/products/rstudio/download/ (accessed on 8 March 2023)), download R Integrated Development Environment (IDE) RStudio. Modify the heat map information using parameters: ”color”, “columnme”, “rownanme”, “cellwidth”, “cellheight”, “cluster_row/col”, “labels_row/col”, etc. included in the package “pheatmap”. The phylogenetic analysis of representative *orthohantavirus* strains was performed using Molecular Evolutionary Genetics Analysis (MEGA) software version 11 by the maximum-likelihood (ML) method and the corresponding optimal model T92 + G + I with the bootstrap value set as 1000 replicates. Orthohantavirus reference sequences used for the comparative analysis: MN022852.1 (Seoul virus), MN022865.1 (Seoul virus), AF288297.1 (Seoul virus), KM948595.1 (Seoul virus), KX079474.1 (Seoul virus), KF977219.1 (Seoul virus), HQ992814.1 (Seoul virus), AF336826.1 (Hantaan virus), KT885047.1 (Hantaan virus), JN831949.1 (Puumala virus), HE801635.1 (Puumala virus), MN639746.1 (Puumala virus), HQ404253.1 (Luxi virus), MK386154.1 (Tula virus), MT514299.1 (Tula virus), MK386155.1 (Tula virus), and DQ825770.1 (Thottapalayam virus).

### 2.12. Statistical Analysis

Epidata software version 3.1 was used to establish the case database and to input the data of the collected medical records, and the software IBM SPSS26 was used for statistical analysis. The enumeration data were expressed by frequency and rate, and the rate was compared with the chi-square test and Fisher exact probability method. When the *p*-value was less than 0.05, the difference was considered statistically significant. Parts of the graphs were made using Excel 2010, and the distribution maps were plotted with the geographic information ArcGIS software version 10.2.

## 3. Results

### 3.1. Cases of HFRS

A total of 190 hospitalized cases were collected, of which laboratory-diagnosed cases accounted for 80.53% (153/190) and clinically diagnosed cases accounted for 19.47% (37/190). The age of onset ranged from 7 to 85 years, with a mean age of 48.5 ± 16.7 years, mainly in the age group of 40–59 years, and the occupation was mainly farmers. Duration of hospitalization (in one year): 95 cases of HFRS were reported from February 2020 to January 2021, including 24.21% (23/95) in spring (January to March); 37.89% (36/95) in summer (April to June); 22.11% (21/95) in autumn (July to September); and 18.95% (18/95) in winter (October to December). Cases were reported throughout the year, and most patients with HFRS were admitted in spring and summer. Distribution of cases in Xiangyun County of Dali Prefecture: Xiangcheng Town accounted for 23.68% (45/190), Yunnan Yi Town accounted for 15.26% (29/190), Xiazhuang Town accounted for 14.74% (28/190), Midian Town accounted for 9.47.% (18/190), Shalong Town accounted for 9.47% (18/190), Pupeng Town accounted for 9.47% (18/190), Liuchang Town accounted for 6.32% (12/190), Hedian Town accounted for 4.74% (9/190), Luming Town accounted for 2.11% (4/190), Dongshan Town accounted for 2.11% (4/190), see Figure 1.

### 3.2. Orthohantavirus Host Animal Survey

Orthohantavirus host animal surveys were conducted in nine towns in Xiangyun County of Dali Prefecture, including Midian Town, Hedian Town, Dongshan Town, Xiangcheng Town, Shalon Town, Liuchang Town, Yunnan Yi Town, Xiazhuang Town, and Pupeng Town (Figure 1). A total of 331 small mammals of 12 species, 4 genera, and 4 families were captured, including 36.25% (120/331) of *Rattus tanezumi*, 24.77% (82/331) of *Apodemus chevrieri*, 16.01% (53/331) of *Rattus norvegicus*, 8.46% (28/331) of *Eothenomys miletus*, 6.34% (21/331) of *Crocidura attenuata*, 2.11% (7/331) of *Tupaia belangeri*, 2.42% (8/331) of *Crocidura dracula*, 1.81% (6/331) of *Mus musculus*, 0.60% (2/331) of *Rattus niviventer*, 0.60% (2/331) of *Eothenomys eleusis*, 0.60% (2/331) of *Mus pahari*, and 0.30% (1/331) of *Rattus nitidus*. The dominant species in residential areas were *Rattus tanezumi* and *Rattus norvegicus*, and the dominant species in field cultivation areas was *Apodemus chevrieri*. After orthohantavirus nucleic acid detection, the total orthohantavirus infection rate in host animals was 2.72% (9/331), among which 6.10% (5/82) were *Apodemus chevrieri*, 3.77% (2/53) were *Rattus norvegicus*, 12.50% (1/8) were *Crocidura dracula*, and 100% (1/1) were *Rattus nitidus* (Table 1).

### 3.3. Clinical Profile of HFRS in Patients

#### 3.3.1. Clinical Manifestations

Among the 190 cases of HFRS patients, fever, rapid onset, and fatigue accounted for 86.84% (165/190), 73.7% (140/190), and 67.89% (129/190), respectively, which were the most common clinical manifestations among hospitalized patients. HFRS has the specific clinical manifestation of “three reds and three pains” (three reds: red face, red neck, and red chest; three pains: headache, back pain, and orbital pain), among which one of three reds accounted for 7.89% (15/190), two of them accounted for 14.21% (27/190), and all three reds accounted for 13.68% (26/190). None of the three reds accounted for 64.2% (122/190). One, two, three, and none in three pains were 36.84% (70/190), 21.58% (41/190), 9.47% (18/190), and 32.11% (61/190), respectively.

On the other hand, among the 190 cases of hospitalized patients with HFRS, the mild, medium, severe, and critical were 36.32% (69/190), 37.37% (71/190), 14.21% (27/190), and 12.11% (23/190), respectively. There were statistically significant differences for the clinical manifestations, fatigue (χ^2^ = 8.71, *p* = 0.033 < 0.05), fever (χ^2^ = 18.79, *p* = 0.000 < 0.05), neck redness (χ^2^ = 10.312, *p* = 0.016 < 0.05), generalized pain (χ^2^ = 8.40, *p* = 0.038 < 0.05), abdominal pain (χ^2^ = 15.42, *p* = 0.001 < 0.05), hypotension, (χ^2^ = 14.54, *p* = 0.002 < 0.05) and shock (χ^2^ = 14.66, *p* = 0.002 < 0.05), in the four clinical subtypes (Table 2).

#### 3.3.2. Laboratory Tests in Hospitalized Patients

Among 190 HFRS cases, the number of leukocytosis cases accounted for 31.58% (60/190), and 53.16% (101/190) of cases showed normal white blood cell counts. Decreased platelet count cases accounted for 65.79% (125/190), and positive urine protein cases accounted for 68.95% (131/190).

#### 3.3.3. Detection of Orthohantavirus RNA and Anti-Orthohantavirus IgM/IgG Antibodies in Hospitalized Patients

The orthohantavirus RNA was detected in 6 of 98 (6.12%) cases with anti-orthohantavirus IgM (+) and IgG (+) patients; 3 of 49 (6.12%) cases with anti-orthohantavirus IgM (+) and IgG (−) patients; 2 of 28 (7.14%) cases with anti-orthohantavirus IgM (−) and IgG (+) patients; and 1 of 15 (6.67%) cases with anti-orthohantavirus IgM (−) and IgG (−) patients. There was no statistically significant difference in the detection rate of orthohantavirus RNA among these four groups (Fisher’s Exact Test Value = 0.466, *p* = 1.000 > 0.05).

#### 3.3.4. Epidemiological Survey of HFRS Patients

A total of 108 questionnaires of epidemiological cases were collected from 190 cases of HFRS patients, of which mild cases were 41.67% (45/108), medium cases were 38.89% (42/108), severe cases were 9.26% (10/108), and critical cases were 10.19% (11/108). There were statistically significant differences between the four clinical subtypes with the presence of rodent activity at home (χ^2^ = 8.75, *p* = 0.031 < 0.05), the presence of rodent-proof equipment in food (χ^2^ = 9.19, *p* = 0.025 < 0.05), and the presence of rodents or rodent excrement in the workplace (χ^2^ = 10.35, *p* = 0.014 < 0.05) (Table 3).

### 3.4. Orthohantavirus Sequence Detection and Similar Comparison

The 21 orthohantavirus-positive samples (12 from HFRS patients and 9 from host animals) were sequenced after RT-PCR amplification. The L fragment size was 400–440 bp, among which primers were designed for the specimen of patient number XYPatient5, and the final sequence of spliced L fragment size was 2089 bp.

At the nucleotide level, orthohantavirus strains of XYPatient5 (GenBank acc. no. OP326502), XYPatient17 (OP326507), XYPatient75 (OP326513) XYPatient10 (OP326504), XYPatient22 (OP326508), from HFRS cases and XYAc15 (OP326534), XYAc16 (OP326535), XYAc17 (OP326536), XYAc19 (OP326537) XYAc20 (OP326538) from *Apodemus chevrieri* in this study shared highest sequence identity, ranging from 94.92% to 99.49%, to the SEOOV GZRn60 strain; orthohantavirus strains of XYPatient35 (OP326509), XYPatient53 (OP326511) from HFRS cases, XYRn5 (OP326533) and XYRn163 (OP326531) from *Rattus norvegicus*, and XYR.nitidus97 (OP326530) from *Rattus nitidus* had highest sequence identity, ranging from 89.00% to 97.46%, to the SEOOV DLR2 strain; orthohantavirus strains of XYPatient7 (OP326503), XYPatient16 (OP326506), XYPatient13 (OP326505), XYPatient62 (OP326512), and XYPatient36 (OP326510) from HFRS cases shared highest sequence identity (58.27–88.47%)to the SEOOV L0199 strain (Figure 2).

The nucleotide sequence identity between XYCd93 (OP326532) from the White-tipped shrew (*Crocidura dracula*) and LUXV LX309 strain from the red-backed vole (*Eothenomys miletus*) was (82.35%) [17]. At the amino acid level, XYCd93 showed 60.29% sequence identity to LX309.

### 3.5. Phylogenetic Analysis of Orthohantavirus Strains Detected in Dali Prefecture

Of the 12 orthohantavirus strains detected in the HFRS patient samples, five orthohantavirus strains (XYPatient5, XYPatient10, XYPatient17, XYPatient22, and XYPatient75) clustered in single clade and were closely related to SEOOV GZRn60 strain, two orthohantavirus strains (XYPatient35 and XYPatient53) were closely related to SEOOV DLR2 strain, and the other five orthohantavirus strains (XYPatient7, XYPatient13, XYPatient16, XYPatient36, and XYPatient62) clustered in a separate clade.

For orthohantavirus strains in host animals, one strain (XYRn5) from *Rattus norvegicus* and five strains (XYAc15, XYAc16, XYAc17, XYAc19, and XYAc20) from *Apodemus chevrieri* were closely related to SEOOV GZRn60 strain. One strain (XYRn163) from *Rattus norvegicus* and one strain (XYR.nitidus97) from *Rattus nitidus* were closely related to SEOOV DLR2 strain. Notably, the orthohantavirus strain (XYCd93) from *Crocidura dracula* and the LU309 strain from red-backed vole (*Eothenomys miletus*) showed a closer evolutionary relationship and clustered together to form a new branch (Figure 3).

## 4. Discussion

Among the clinical manifestations of HFRS in this study, fatigue, fever, and redneck were more common in mild and medium cases. For severe and critical patients accompanied by some basic diseases or delayed treatment, which will lead to aggravation of the disease, the patient is more prone to systemic pain, diarrhea, abdominal pain, hypotension, and shock in severe cases. In addition, HFRS has specific “three reds and three pains” in the clinic; 122 of 190 cases did not have “three reds” of clinical manifestations, and 61 cases did not have “three pains” of clinical manifestations. Therefore, the specific clinical manifestations of HFRS for diagnosis were not comprehensive, meaning that it is easy to miss the diagnosis and that further laboratory testing is warranted. Clinical laboratory tests are usually considered the leading indicators, such as increased white blood cell count, decreased platelet count, and positive urine protein in patients with HFRS. In this survey, 53.15% of patients with HFRS had normal white blood cell counts; therefore, white blood cell counts, platelet counts, and urine protein do not directly reflect whether a patient has HFRS, and the accuracy of laboratory indicators for HFRS can be affected by the presence of certain underlying diseases in patients. Studies have shown that the levels of procalcitonin (PCT) and C-reactive protein (CRP) in HFRS patients are significantly increased and are positively associated with the clinical severity of HFRS [27,28]. Therefore, the clinical diagnosis of HFRS combined with PCT and CRP results can improve the accuracy of diagnosis. Immunological tests now usually use colloidal gold test cards for orthohantavirus antibody detection in China, which can rapidly detect the presence of anti-orthohantavirus-specific IgM and IgG antibodies in patients’ serum samples and help clinicians to confirm the diagnosis of HFRS. Among 190 cases of HFRS, the positive results of the orthohantavirus antibody test for IgM antibodies suggested that the patients had a recent infection, accounting for 77.37% (147/190). Considering the above laboratory test results together, it is important to diagnose HFRS and accurately determine the disease’s severity. The RNA of orthohantavirus was tested in the sera of 190 patients with HFRS, and the positive rate was 6.32% (12/190). The low detection rate of the RNA was mainly due to the immunological diagnosis of orthohantavirus antibody-positive cases; the possible reason was that the patients had passed the viremia stage and the virus content in the serum was low, and the RNA could not be detected. In addition, orthohantavirus is an RNA virus, and RNA is difficult to preserve and easy to decompose. Several factors may lead to the low detection rate of orthohantavirus RNA.

Our present study indicated that fewer people in the natural foci had been vaccinated against HFRS in the past. For residents in residential areas, rodent control facilities with rodent-infested food at home are inadequate, or they often work in wild farms or fields, and have direct or indirect contact with rodent excreta or rodent-contaminated food that are more likely to increase the risk of HFRS. Due to the limited sample size of HFRS patients, there may be some mild patients not hospitalized and treated, or patients choose other hospitals far away. The investigation of symptomatic cases of 108 hospitalized HFRS patients could not accurately conclude the overall epidemiology of HFRS in Xiangyun County of Dali Prefecture, and further investigation is required.

HFRS in China is mainly caused by HTNOV, SEOOV, and PUUOV [29]. Yunnan Province is one of the main epidemic areas of HFRS in China, dominated by HTNOV and SEOOV. In this study, the genotype distribution of patients infected with orthohantavirus in Xiangyun County of Dali Prefecture, Yunnan Province, was studied, and the results showed that the nucleotide sequence identity between 12 strains of HFRS patients in the L partial segment was 78.37%~98.47%, all of which were SEOOV genotype orthohantavirus, and the L fragment sequence analysis of orthohantavirus showed high identity and low variation. Thus, it is presumed that orthohantavirus infection in the HFRS epidemic area of Xiangyun County of Dali Prefecture is mainly the SEOOV genotype.

Among the orthohantavirus host animals captured from Xiangyun County of Dali Prefecture, the dominant species of rodents indoors is *Rattus norvegicus*, and the dominant species in the field is *Apodemus chevrieri*. XYCd93 is an orthohantavirus strain isolated from a *Crocidura dracula* captured in Dasongping Village, Midian Town, and its BLASTn results are most homologous to the PUUOV Osnabruck M34 strain in Germany [30]. The nucleotide identity comparison results of the most conserved part of the L fragment show that XYCd93 has only 75.96% with the Osnabruck M34 strain, which may be a new orthohantavirus and needs further verification. The host animals harboring diverse genotypes of orthohantavirus with genetic diversity are different from the orthohantavirus genotypes in HFRS patients, which only contain SEOOV. The plausible reason is likely because the dominant species in residential areas is *Rattus norvegicus* carrying SEOOV. Humans have higher chances of contacting *Rattus norvegicus* and are more likely to be infected by SEOOV-causing HFRS.

The incidence of HFRS is influenced by various factors, including socioeconomic conditions, climatic factors, and rodent-human contact [31]. For the natural focal disease of HFRS in Yunnan Province, improving the sanitary conditions of people living and working places, reducing rodent density, personal protection, and vaccination are the key measures to reduce the cases of HFRS [32]. Molecular epidemiological surveillance of orthohantavirus can effectively delineate and manage the epidemic source sites. By combining the publicity and education of the community and social health, we can improve the awareness of the population on the transmission route of the disease, enhance the awareness of prevention, achieve early detection and diagnosis, and the hospital can perform a targeted diagnosis and treatment. By improving the corresponding diagnosis and treatment means, the detection rate of HFRS can be increased, and the fatality rate of HFRS can be reduced.

## 5. Conclusions

In conclusion, this study on the clinical and molecular epidemiology of HFRS in Dali Prefecture of Yunnan Province provides important scientific insights into the disease severity, environment correlation, and prevalence of orthohantavirus infection. Furthermore, we determined the infection rate and genetic diversity of orthohantaviruses in humans and various animal hosts. Our findings will contribute to the understanding of the interplay of orthohantavirus, humans, and animal hosts, highlighting the need for robust surveillance, improved sanitation conditions, and public health awareness to minimize the risk of zoonotic transmission of orthohantavirus in the HFRS natural foci area.

## Figures and Tables

**Figure 1 vaccines-11-01477-f001:**
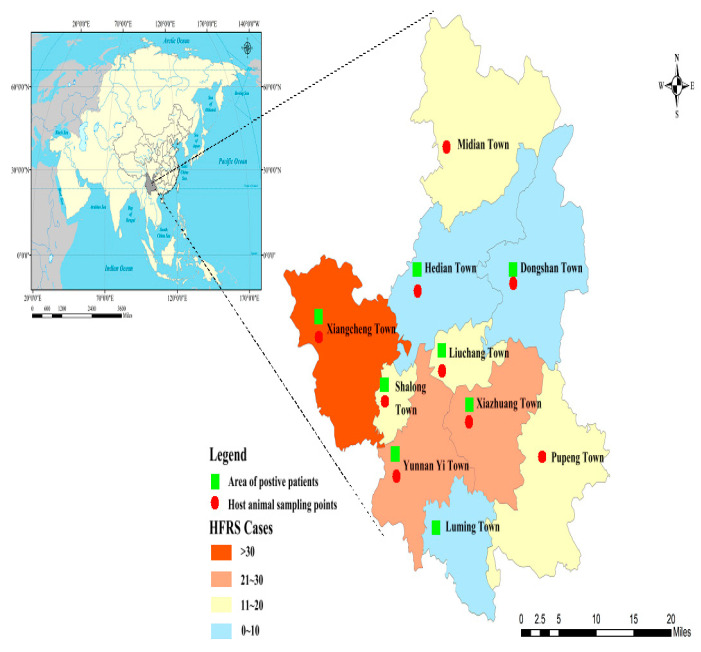
The orthohantavirus infection rates and distribution in HFRS patients and host animals in Xiangyun County, Dali Prefecture, Yunnan Province, China.

**Figure 2 vaccines-11-01477-f002:**
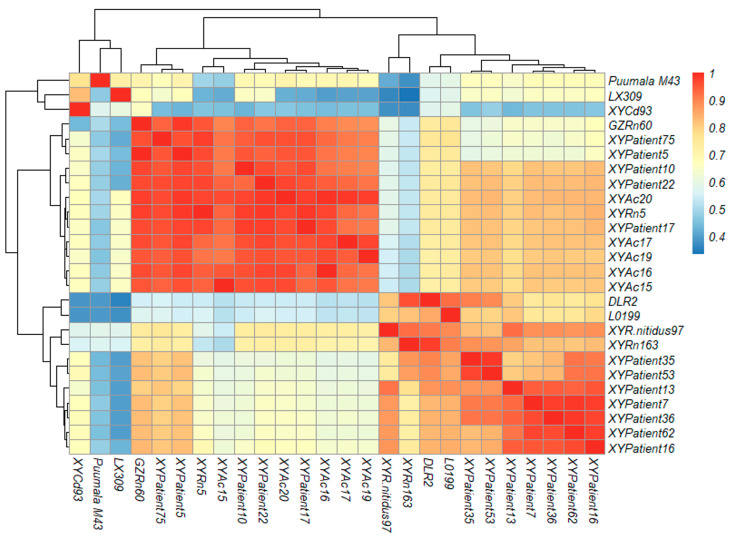
Identity comparisons of nucleotide and amino acid sequence of orthohantavirus partial L fragments in HFRS patients and rodents in this study. Note: Top right is nucleotide sequence homology, bottom left is amino acid sequence homology.

**Figure 3 vaccines-11-01477-f003:**
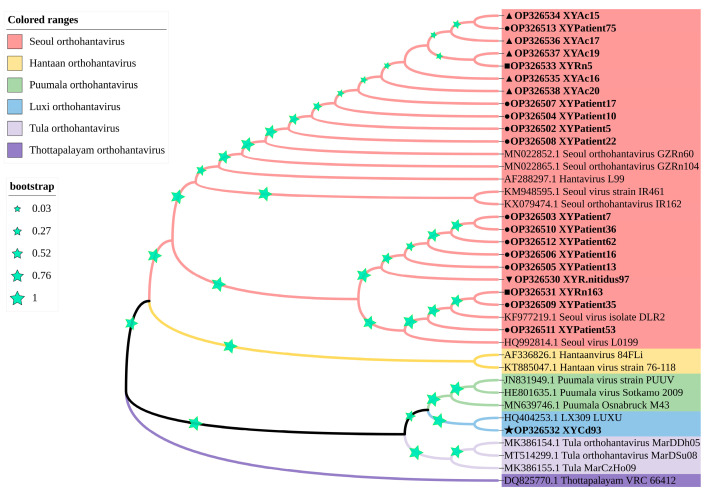
Phylogenetic tree of orthohantavirus partial L fragment in HFRS patients and host animals in Xiangyun County, Dali Prefecture, Yunnan Province, China. ● indicates orthohantavirus strains detected in HFRS patients; ▲ indicates orthohantavirus strains detected in *Apodemus chevrieri*; ■ indicates orthohantavirus strains detected in *Rattus norvegicus*; ★ indicates orthohantavirus strains detected in *Crocidura dracula*; ▼ indicates orthohantavirus strains detected in *Rattus nitidus*.

**Table 1 vaccines-11-01477-t001:** Orthohantavirus infection rates in host animals in Xiangyun County, Dali Prefecture, Yunnan Province, China.

Species	Location									Total
	Xiangcheng	Shalong	Yunnanyi	Midian	Hedian	Pupeng	Liuchang	Xiazhuang	Dongshan	
*Rattus norvegicus*	14.29% (1/7)	0 (0/3)	0 (0/5)	0 (0/3)	3.45% (1/29)	0 (0/1)	0 (0/2)	0 (0/3)		3.77% (2/53)
*Rattus tanezumi*	0 (0/15)		0 (0/24)	0 (0/14)	0 (0/56)		0 (0/2)	0 (0/4)	0 (0/5)	0 (0/120)
*Apodemus chevrieri*	16.67% (5/30)	0 (0/13)	0 (0/20)	0 (0/5)	0 (0/14)					6.10% (5/82)
*Crocidura attenuata*	0 (0/1)	0 (0/2)	0 (0/1)	0 (0/1)	0 (0/16)					0 (0/21)
*Eothenomys miletus*	0 (0/2)	0 (0/1)	0 (0/4)	0 (0/2)	0 (0/17)		0 (0/1)	0 (0/1)		0 (0/28)
*Tupaia belangeri*	0 (0/1)			0 (0/3)	0 (0/2)					0 (0/7)
*Rattus niviventer*	0 (0/1)								0 (0/1)	0 (0/2)
*Crocidura dracula*	0 (0/2)	0 (0/3)	0 (0/1)	50.00% (1/2)						12.50% (1/8)
*Eothenomys eleusis*					0 (0/2)					0 (0/2)
*Mus pahari*					0 (0/2)					0 (0/2)
*Rattus nitidus*					100% (1/1)					100% (1/1)
*Mus musculus*					0 (0/6)					0 (0/6)
Total	59	22	55	30	145	1	5	8	6	2.72% (9/331)

**Table 2 vaccines-11-01477-t002:** Clinical performance of 190 cases with HFRS in Xiangyun County, Dali Prefecture, Yunnan Province, China.

Symptoms and Signs	Mild	Medium	Severe	Critical	Total	χ^2^	*p*-Value
Rapid onset	44 (63.77%)	56 (78.87%)	21 (77.78%)	19 (82.61%)	140 (73.68%)	5.66	0.129
Lack of power	39 (56.52%)	49 (69.01%)	22 (81.48%)	19 (52.61%)	129 (67.89%)	8.71	0.033
Fever	52 (75.36%)	70 (98.59%)	25 (92.59%)	18 (78.26%)	165 (86.84%)	18.79	0.000
Headaches	36 (52.17%)	46 (64.79%)	10 (37.04%)	14 (60.87%)	106 (55.79%)	6.79	0.079
Back Pain	25 (36.23%)	27 (38.03%)	14 (51.85%)	9 (39.13%)	75 (39.47%)	2.09	0.552
Eye socket pain	6 (8.70%)	12 (16.90%)	4 (14.81%)	3 (13.04%)	25 (13.16%)	2.14	0.544
Blush	17 (24.64%)	23 (32.39%)	12 (44.44%)	8 (34.78%)	60 (31.58%)	3.74	0.291
Neck Red	15 (21.74%)	23 (32.39%)	15 (55.56%)	7 (30.43%)	60 (31.58%)	10.31	0.016
Chest Red	6 (8.70%)	12 (16.90%)	6 (22.22%)	3 (13.04%)	27 (14.21%)	3.59	0.309
Joint pain	9 (13.04%)	7 (9.86%)	3 (11.11%)	1 (4.35%)	20 (10.53%)	1.44	0.696
Whole Body	27 (39.13%)	45 (63.38%)	15 (55.56%)	12 (52.17%)	99 (52.11%)	8.4	0.038
Abdominal pain	18 (26.09%)	7 (9.86%)	3 (11.11%)	10 (43.48%)	38 (20.00%)	15.42	0.001
Diarrhea	9 (13.04%)	7 (9.86%)	6 (22.22%)	4 (17.39%)	26 (13.68%)	2.84	0.417
Constipation	2 (2.90%)	3 (4.23%)	0 (0.00%)	0 (0.00%)	5 (2.63%)	2.07	0.557
Disgusting	26 (37.68%)	23 (32.39%)	12 (44.44%)	13 (56.52%)	74 (38.95%)	4.66	0.199
Vomiting	19 (27.54%)	10 (14.08%)	7 (25.93%)	7 (30.43%)	43 (22.63%)	4.88	0.181
Conjunctival congestion	16 (23.19%)	25 (35.21%)	12 (44.44%)	7 (30.43%)	60 (31.58%)	4.76	0.190
Puffy eyelids	3 (4.35%)	1 (1.41%)	0 (0.0%)	2 (8.70%)	6 (3.16%)	4.22	0.239
Little or no urination	13 (18.84%)	19 (26.76%)	6 (22.22%)	6 (26.09%)	44 (23.16%)	1.37	0.714
Low blood pressure	8 (11.59%)	10 (14.08%)	8 (29.63)	10 (43.48%)	36 (18.95%)	14.54	0.002
Shock	4 (5.80%)	5 (7.04%)	6 (22.22%)	7 (30.43%)	22 (11.58%)	14.66	0.002

**Table 3 vaccines-11-01477-t003:** Epidemiological findings of HFRS inpatients in Xiangyun County, Dali Prefecture, Yunnan Province, China.

Survey Items	Mild	Medium	Severe	Critical	Total	χ^2^	*p*-Value
Previous history of HFRS	4 (8.89%)	2 (4.76%)	0 (0.00%)	0 (0.00%)	6(5.56%)	1.17	0.845
Have been vaccinated against HFRS	2 (4.44%)	0 (0.00%)	1 (10.00%)	0 (0.00%)	3 (2.78%)	3.81	0.233
Family members have experienced similar symptoms	1 (2.22%)	0 (0.00%)	0 (0.00%)	0 (0.00%)	1 (0.93%)	2.75	1.000
There are rat activities at home	19 (42.22%)	24 (57.14%)	8 (80.00%)	9 (81.81%)	60 (55.56%)	8.75	0.031
With rodent-proof equipment for grain belt	7 (15.56%)	11 (26.19%)	6 (60.00%)	4 (36.36%)	28 (25.93%)	9.19	0.025
All rats or mouse excreta in the workplace	10 (22.22%)	13 (30.95%)	6 (60.00%)	7 (63.63%)	36 (33.33%)	10.35	0.014
Exposure to rats within one month before the onset	15 (33.33%)	15 (35.71%)	4 (40.00%)	2 (18.18%)	36 (33.33%)	1.44	0.704
History of going out in the two months before the onset of the disease	3 (6.67%)	3 (7.14%)	1 (10.00%)	0 (0.00%)	7(6.48%)	1.01	0.874

## Data Availability

All the orthohantavirus sequences derived in this study can be obtained and downloaded from NCBI database (https://www.ncbi.nlm.nih.gov (accessed on 7 May 2023) via the following accession numbers: XYPatient5 (OP326502), XYPatient7 (OP326503), XYPatient10 (OP326504), XYPatient13 (OP326505), XYPatient16 (OP326506), XYPatient17 (OP326507), XYPatient22 (OP326508), XYPatient35 (OP326509), XYPatient36 (OP326510), XYPatient53 (OP326511), XYPatient62 (OP326512), XYPatient75 (OP326513), XYR.nitidus97 (OP326530), XYRn163 (OP326531), XYCd93 (OP326532), XYRn5 (OP326533), XYAc15 (OP326534), XYAc16 (OP326535), XYAc17 (OP326536), XYAc19 (OP326537), and XYAc20 (OP326538).

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
