# Peer review of "Clinical and Molecular Epidemiology of Hemorrhagic Fever with Renal Syndrome Caused by Orthohantaviruses in Xiangyun County, Dali Prefecture, Yunnan Province, China"

_vaccines, 2023, doi:10.3390/vaccines11091477_

Round 1

Reviewer 1 Report

The manuscript entitled “Clinical and Molecular Epidemiology of Hemorrhagic Fever with Renal Syndrome Caused by Orthohantaviruses in Xiangyun County, Dali Prefecture, Yunnan Province, China” was generally well-written, and provided additional knowledge of the genetic diversity of orthohantavirus in local area. Because of local focusing, this study contained a weak point. However, in my opinion, that could be improved by (1) additional genetic/ phylogenetic analyses with a larger dataset which are from other countries, and (2) broaden the current discussion with more similar studies in the world. Beside of that, I have the following comments to the manuscript.

Abstract:

It was quite long. Some information from lines 21-27, for example might not necessary.

Introduction:

- More citations are required to make supports for information given in lines 53-64.

- Line 60: incorrect spelling of “othohantavirus”.

- Line 72: “at least 10 HTNOV subtypes and 6 SEOOV subtypes”. However, looking at the cited reference [4], the original publication referred to “nine” and “five” subtypes and some not-assigned subtypes. Please verify and explain the differences.

Materials and Methods

- Wrong spelling of “HFPS” in lines 108, 128.

- Lines 129-130: it was stated that the patients’ sera were collected from October 2020. By looking at the information given in GenBank (for example XYPatient5, OP326502), the collection date was "03-Jul-2019". Please verify and make a consistency.

- Lack of the method for the result mentioned in sub-section 3.3.2 (white blood count, urine protein) and 3.3.3 (detection of IgG/ IgM antibodies).

- Lack of the method to draw figure 2.

Results

- Figure 1: (1) the color code for Luming Town should be similar to those of Hedian Town and Dongshan Town; (2) this is a scientific paper, please remove the dashed lines in the inserted map.

- Table 1: to make a consistency, please round the decimal numbers down to the nearest tenth. For example: line 36 of the abstract it was “3.8% (2/53)”, table 1 was “3.77% (2/53)”.

- Figure 2: The leaves of the left and upper dendrograms were not similar. Please take a note in the figure caption.

- Figure 3: it would be better if known HTNOV and SEOOV subtypes from reference [4] were included for phylogenetic inference.

Author Response

Please see the attachment below. Thank you!

Author Response

(The authors gave the same response as above.)

Round 2

Reviewer 1 Report

There are still some points which were not corrected in this revision:

- Point 8: “Sequence similarity heatmaps are plotted using R and its packets.”: please give more details or reference or link to help audience who is interested in reproduction of the result. Spelling of “packets” was incorrect.

- Point 10: the dashed lines were remained. Please remove.

- Point 11: it was still observed “3.77% (2/53)” in Table 1.

Author Response

Reviewer 1(Round 2)

- Point 8: “Sequence similarity heatmaps are plotted using R and its packets.” please give more details or reference or link to help audience who is interested in reproduction of the result. Spelling of “packets” was incorrect.

Response 8:  Changed “packets” to “package”.

Added: On the official website of R (https://www.r-project.org/), select the corresponding operating system (Windows, Mac, and Linux) to download the latest installation package version, and follow the installation wizard to complete the software installation. By website (https://www.rstudio.com/products/rstudio/download/), download R Integrated Development Environment (IDE) RStudio. Modify the heat map information using parameters:"color", "columnme", "rownanme", "cellwidth", "cellheight", "cluster_row/col", "labels_row/col", etc. included in the package “pheatmap” .

Revised, please see the text.

- Point 10: the dashed lines were remained. Please remove.

Response 10: Revised, please see Figure1

- Point 11: it was still observed “3.77% (2/53)” in Table 1.

Response 11: The relevant numbers in this manuscript are retained to two digits after the decimal point, and for percentages, the decimal part after the percentile symbol retained 2 digits Therefore it was still observed "3.77% (2/53)" in Table 1. The same is true for other relevant figures, please see the text.

Round 3

Reviewer 1 Report

I do not have any more specific comment. Just double check spelling for the entire manuscript to avoid error, such as "packag".

Author Response

We have checked spelling for the entire manuscript. Revised, please see the text. Thank you very much.
